# Application of Machine Learning and Deep Neural Visual Features for Predicting Adult Obesity Prevalence in Missouri

**DOI:** 10.3390/ijerph21111534

**Published:** 2024-11-19

**Authors:** Butros M. Dahu, Carlos I. Martinez-Villar, Imad Eddine Toubal, Mariam Alshehri, Anes Ouadou, Solaiman Khan, Lincoln R. Sheets, Grant J. Scott

**Affiliations:** 1Institute for Data Science and Informatics, University of Missouri, Columbia, MO 65211, USA; sknnh@missouri.edu (S.K.); sheetslr@missouri.edu (L.R.S.); scottgs@missouri.edu (G.J.S.); 2Department of Health Management and Informatics, University of Missouri, Columbia, MO 65211, USA; 3Department of Electrical Engineering and Computer Science, University of Missouri, Columbia, MO 65211, USA; cmartinezvillar@mail.missouri.edu (C.I.M.-V.); itoubal@missouri.edu (I.E.T.); msalshehri@missouri.edu (M.A.); aomqc@missouri.edu (A.O.)

**Keywords:** DCNN, geospatial, machine learning, obesity rate, satellite imagery

## Abstract

This research study investigates and predicts the obesity prevalence in Missouri, utilizing deep neural visual features extracted from medium-resolution satellite imagery (Sentinel-2). By applying a deep convolutional neural network (DCNN), the study aims to predict the obesity rate of census tracts based on visual features in the satellite imagery that covers each tract. The study utilizes Sentinel-2 satellite images, processed using the ResNet-50 DCNN, to extract deep neural visual features (DNVF). Obesity prevalence data, sourced from the CDC’s 2022 estimates, is analyzed at the census tract level. The datasets were integrated to apply a machine learning model to predict the obesity rates in 1052 different census tracts in Missouri. The analysis reveals significant associations between DNVF and obesity prevalence. The predictive models show moderate success in estimating and predicting obesity rates in various census tracts within Missouri. The study emphasizes the potential of using satellite imagery and advanced machine learning in public health research. It points to environmental factors as significant determinants of obesity, suggesting the need for targeted health interventions. Employing DNVF to explore and predict obesity rates offers valuable insights for public health strategies and calls for expanded research in diverse geographical contexts.

## 1. Introduction

This research study addresses the escalating obesity epidemic in the United States and globally, emphasizing its complex relationship among various factors. It highlights the significant advancements in understanding this issue through deep learning and machine learning techniques, particularly the analysis of large satellite imagery datasets (commonly called Earth Observation). It also focuses on the discrepancies in previous studies and introduces a novel methodology using deep convolutional neural networks (DCNN) to extract features from satellite images, combined with an advanced machine learning regression model to predict obesity prevalence from satellite images. This approach aims to provide a comprehensive and scalable analysis of the correlation between deep neural visual features (DNVFs) and obesity rates utilizing publicly accessible data.

Obesity has escalated to an epidemic level across the United States [1]. From 2001 to 2018, there has been a steady rise in obesity rates among adults over 20 years old [1]. The Global Burden of Disease study reveals that over 603 million adults were obese globally in 2015 [2,3]. In the United States, over a third of adults are obese, with 46 states reporting an adult obesity rate of at least 25% [2,4]. Obesity is a complex issue influenced by various factors such as the higher risk of various non-communicable diseases [5,6], including cancer, mobility issues, heart disease, mental health problems, osteoarthritis, sleep apnea, stroke, and Type 2 diabetes [1,2,7].

Additionally, obesity causes an estimated 300,000 deaths annually, just behind smoking-related deaths [1,8]. Unhealthy eating habits and inactive lifestyles are often linked to aspects of both the social and built environments, encompassing natural and modified physical surroundings [7]. It impacts health by dictating the availability of resources like housing, spaces for activities and recreation, and community design features [2,9,10,11,12].

The growing prevalence of obesity and related health issues in urban settings has led researchers to delve into the complex relationship between public health and individuals’ surroundings, which makes using raw overhead satellite images to predict a health metric an interesting approach [13,14,15,16]. This exploration has been significantly advanced by the application of deep learning and machine learning techniques, which have enabled the analysis of large datasets from diverse sources such as satellite and street view imagery [17,18,19,20,21].

Previous research indicates a connection between various aspects of the built environment and their impact on obesity and physical activity throughout different phases of life [7,22,23,24]. Previous studies also highlight a link between obesity and several environmental variables such as the walkability of an area, its land use patterns, the extent of urban sprawl, the type of residential area, availability of resources like recreational facilities and food establishments, the degree of socio-economic deprivation, and the level of perceived safety in an area [25,26,27,28]. Moreover, being close to and having access to natural spaces and sidewalks is associated with higher and more consistent physical activity levels, particularly in urban environments [29,30,31].

Although there is a recognized relationship between obesity and the built environment, there have been noted discrepancies in the findings of different studies and across various geographical areas regarding the impact of specific built environment features on obesity rates [25,32,33,34]. These inconsistencies might stem from differences in the methods and tools used for measurement in these studies, which complicates the evaluation and comparison of the results [35,36,37]. Additionally, measuring these environmental features often involves significant costs and time, and is prone to human error and bias [38,39,40]. There is a need for methodologies that offer uniform measurement standards to facilitate comparisons across different studies [36,41,42]. Accurately determining the influence of the built environment on obesity is important for the development and execution of effective community-based prevention and intervention strategies [38,43,44,45].

Herein, we present a novel methodology to thoroughly investigate the link between the prevalence of adult obesity using solely overhead satellite images. This method utilizes a deep learning technique, specifically a DCNN is used to analyze the physical characteristics of neighborhoods from medium-resolution (10 m) satellite images. This approach builds on the work of Maharana et al. [2] and Nguyen et al. [46], who employed DCNNs to categorize images from Google Street View, examining the relationship between obesity and specific elements like crosswalks, building types, and the presence of greenery or landscaping. However, their research was constrained to these three preselected features and did not fully leverage the potential of DCNNs to identify features correlated with obesity rates autonomously. Our method, in contrast, offers a more comprehensive analysis of overhead image features, pinpointing detailed correlations with obesity rates at the census tract level in 1052 census tracts across the State of Missouri in the United States. Additionally, our approach is scalable, utilizes publicly accessible data and computational resources, facilitates comparison between studies, and can be applied to different geographic sites and regions.

The objective of this study is to employ deep convolutional neural networks (DCNNs) to analyze medium-resolution satellite images and identify physical characteristics of neighborhoods that correlate with obesity rates. By focusing on the 1052 census tracts in Missouri, USA, the study offers a more comprehensive and scalable approach to assessing the impact of deep neural visual features on obesity prevalence, enhancing the precision of public health interventions and policy-making. The importance of predicting obesity using deep neural network features is underlined by our method’s ability to enhance the precision of public health interventions and improve the granularity of epidemiological studies.

This paper is structured as follows. In Section 2, we provide an overview of the related literature. Section 3 outlines the methods employed in this study, followed by a discussion of the results in Section 4. In Section 5, we discuss the findings’ implications and acknowledge our work’s limitations. Finally, Section 6 presents the overall conclusions and key takeaways.

## 2. Related Work

Songhyeon SH et al. [47] and Lam TM et al. [48] showcase the diverse methodologies employed in the field. They used a combination of multinomial logistic regression and Deep Neural Network algorithms to analyze community health survey data from 2018 to 2020. The study underscores integrating statistical methods with advanced machine learning techniques in public health research. Lam TM et al. [48] conducted an umbrella review, synthesizing evidence from 32 systematic reviews to evaluate the connection between the built environment and obesity. This meta-analysis focused on the methodological aspects of the reviewed studies, particularly assessing biases and thematic areas, thus providing a comprehensive overview of research trends and methodological approaches in this domain.

The studies by Alkhalaf M et al. [49] and An R et al. [50] delve into the advanced applications of machine learning and artificial intelligence in obesity research. Alkhalaf M et al. conducted a comprehensive analysis of ML applications in adult obesity studies, focusing on the evaluation of various algorithms, including regression models, neural networks, and deep learning techniques [49]. R. An et al. assessed the technical aspects of AI models, particularly ML and DL, in obesity research, analyzing 46 studies that utilized a range of AI methodologies [50]. Zhou et al. [19,51] explored the use of machine learning models in the context of obesity research, with a focus on biomarker detection and intervention strategies. Their review categorized ML models into supervised and unsupervised learning types and detailed 25 open-source ML algorithms, platforms, and databases relevant to various aspects of obesity research.

DCNNs have seen emerging use in population health studies, across various image and sensing modalities and tasks [52,53,54]. Maharana et al. study utilized the VGG-CNN-F model to analyze approximately 150,000 high-resolution satellite images [2]. Newton et al. [55] employed the Xception DCNN architecture, allowing for a more efficient process than traditional DCNNs. Yue X et al. [56] and Phan L et al. [57] both emphasized the use of CNNs for analyzing neighborhood characteristics, but with different approaches and scales. They employed VGG19 [58] and ResNet18 [59] architectures to analyze a vast dataset of 164 million Google Street View images. The study highlighted the effectiveness of using different CNN architectures to manage and interpret large-scale image data. Phan L et al. [57], on the other hand, focused on a dataset of 31,247,167 Google Street View images, utilizing the VGG-16 model. Their objective was to evaluate built environment indicators at the state level in the United States and explore their association with public health outcomes.

## 3. Materials and Methods

The study analysis consisted of three steps. First, we processed Sentinel-2 satellite images to extract features of any environment using ResNet-50 [59]. Second, we merged the census tract polygons from Tiger Line with the polygons from the CDC data to match each census tract with its respective obesity rate [60]. Third, we used a machine learning model, generalized linear model (GLM), random forest, and 10-fold-cross validation to build a parsimonious model to predict the obesity rate for each census tract and to assess the association between the built environment and obesity prevalence. The study was exempt from institutional review board approval because this research used existing data and records collected by external parties in such a manner that individuals cannot be identified.

Figure 1 summarizes the methodology of our research study, which leverages remote sensing and machine learning to estimate regional obesity rates. Starting with medium-resolution images from Sentinel-2, the study utilizes the ResNet-50 to extract pertinent DNVF from the satellite data. These features, encoded as a 2048-dimensional vector or concepts embedding, contain critical visual information that reflects the environment’s characteristics, potentially linked to obesity prevalence.

The extracted features then undergo regression analysis within a machine learning framework to predict the estimated obesity rates for the study area. This statistical approach models the relationship between the DNVF and the obesity rates, providing an innovative angle to public health research.

### 3.1. Obesity Prevalence Data

We utilized 2022 estimates of annual crude obesity prevalence at the census tract level, derived from the 500 Cities project (PLACES: Local Data for Better Health, accessed on 17 March 2024). These estimates are based on data from the Behavioral Risk Factor Surveillance System, which surveys individuals aged 18 and older. Obesity is identified using a body mass index (BMI) threshold of 30, calculated as the individual’s weight in kilograms divided by their height in meters squared [61]. Our study focused on the Mid-Missouri region in the United States. The 1052 census tracts (Missouri State) covered in this study have an aggregate area of 69,707 square miles (180,540 square km). They have a total population of 6.2 million (based on 2020 census).

The CDC data lists 1387 census tracts, 4506 block groups, and 343,565 census blocks. Given that the number of Tiger Line census tract shapefiles in the state was 1654, polygon (census tract) IDs in the Tiger Line data had to be aligned to census tract IDs in the CDC data. To fix the mismatching issue, census tracts with subdivisions (tract names with two trailing digits different from zero) in both datasets were joined into larger polygons.

First, we joined the polygons in the Tiger Line dataset by removing all subdivisions. Of the initial 1654 polygons, 914 had names following a naming convention of the type “XXXX.YY”, where the “YY” corresponded to the subdivision within a particular tract. The remaining 740 had no subdivisions and names containing two trailing zeros (e.g., “XXXX.00”). Setting the two trailing digits to zero in these 914 names resulted in repeated names with 323 unique names. If a name was repeated, all repeated elements were joined to become a single polygon, which resulted in a set of 1063 polygons.

Similarly, the IDs of the census tracts in the CDC data consisted of a string of digits, with the last two digits corresponding to a set of subdivisions different from that of the Tiger Line data. Of the 1387 tracts, 881 had IDs with two trailing zeros (no subdivisions), while the remaining 506 had subdivisions. We set the initial 506 string IDs to a single subdivision with repeated entries, which resulted in 178 unique IDs. Since each of these new unique IDs had multiple obesity rates, an average obesity rate was calculated as a proxy for the obesity rate of the newly joined area. The average obesity rate for these joint areas, weighted by census tract population, was calculated as
(1)$$\bar{w}=\frac{\sum_{i=1}^{n}w_{i}x_{i}}{\sum_{i=1}^{n}w_{i}},$$
where $\bar{w}$ is the new obesity rate for the joint area, *n* is the number of subdivisions (repeated entries) within the original census tract, $w_{i}$ is the population in a subdivision *i*, and $x_{i}$ is the obesity rate in *i*. The joined 1059 CDC entries were then matched to the joined 1063 Tiger Line tracts, for a final overlapping set of 1055 census tract polygons with their corresponding obesity rates, which we used as the inputs to our models.

### 3.2. Acquiring Satellite Imagery

We determined our image inputs by selecting the Sentinel-2 products intersecting our previously defined set of census tracts. These satellite products were downloaded from ESA’s Copernicus Dataspace Ecosystem. Since ESA’s OpenSearch API uses HTTP requests to search for products, we defined a shortened geometry string that could fit in our search query. This was done by joining our previously defined census tract polygons into a state boundary that was further simplified into a closed polygon of 54 vertices using the implementation of the Douglas-Pecker algorithm included in GeoPandas.

Our search resulted in 187 intersecting Sentinel-2 products between 1 July 2022, and 31 August 2022. Overlapping images were removed in two steps. First, products with completely overlapping geometries (corresponding to the same UTM zone tile) were filtered by discarding all but the product with the largest area and the lowest cloud percentage. Second, seven partially overlapping products (which also happened to have little state coverage) were discarded after visual inspection. This resulted in a set of 33 Sentinel-2 images that were used to define our inputs to the neural network. These products were downloaded from ESA’s Dataspace Ecosystem to Nautilus. All 33 Sentinel-2 image sizes were 10,980 by 10,980 pixels. The images were then normalized to values between 0 and 1 and cropped into chips of 224 by 224 pixels. This created a total of 82,500 three-band (RGB) image chips.

### 3.3. Image Processing

DCNNs (e.g., ResNet-50) have made significant strides in various computer vision tasks such as object detection and image segmentation, especially when dealing with extensive data sets [3,62]. These advances are not only crucial in general technology fields but also have profound implications in healthcare, such as in the identification of skin cancer, and social issues like poverty estimation [4,63]. Due to the absence of a substantial labeled satellite dataset–with sufficient data to train a neural network–for categorizing regions with high and low obesity rates, we used the DNVF from a pre-trained neural network to obtain the built area features of the 82,500 satellite image chips. We intersected our 82,500 chips with our Missouri census tracts which results in 63,592 chips that were usable. Only chips that intersected census tracts and were fully within the joint polygon of the state of Missouri were and everything else was discarded.

We used the ResNet-50 network, which is composed of 50 layers (48 convolution layers along with one max pool and one average pool layer) and is trained on approximately 1.2 million images from the ImageNet database (a data set of >14 million images used for large-scale visual recognition challenges) [64] for recognizing objects belonging to 1000 categories [64]. For each chip we passed through the network, we extracted the 2048 features from the last hidden layer of the network before the output layer [65]. Before passing through, each image is standardized using the ImageNet per-band mean and standard deviation. This resulted in a final dataset–used for our machine learning analysis of 1051 rows and 2048 columns, with each row corresponding to the average feature vector of a census tract. Since each census tract in our dataset could have more than one image chip intersecting it, we calculated a corresponding weighted mean feature vector for a tract with the features of the intersecting chips [66]. For a census tract *t*, its mean feature vector $F_{t}$ was calculated as:(2)$$F_{t}=\frac{\sum_{\forall c\in C}w_{c}F_{c}}{\sum_{\forall c\in C}w_{c}},$$
where C is the set of image chips intersecting the census tract, $F_{c}$ is a ResNet-50 feature vector obtained from chip *c*, and $w_{c}$ is the (scalar) number of pixels in *c* intersecting the census tract.

To include the obesity rate of a given chip in our analysis, the inverse case was considered. When a chip intersected with more than one census tract, we calculated the obesity rate in such a chip as the average obesity rate of all the intersecting census tracts [66]. So, for a chip *c* the weighted obesity rate is similarly calculated as:(3)$$o_{c}=\frac{\sum_{\forall t\in T}w_{t}o_{t}}{\sum_{\forall t\in T}w_{t}},$$
where T is the non-empty set of tracts intersecting a chip *c*, $o_{t}$ is the obesity rate in a census tract, and $w_{t}$ is the number of pixels in *c* corresponding to the *t* census tract. We do not link these features to specific elements in the built environment or the obesity prevalence. Rather, these DNVFs collectively represent an indicator to help predict the obesity prevalence for each census tract.

The image chips were cropped for a size of 224 pixels by 224 pixels or 2240 m (2.24 km) by 2240 m (2.24 km). The GPU-accelerated processing speed was approximately 77.51 tiles per second, for a total of 17.73 min. This translates to a rate of 0.0129 s per tile.

Figure 2 shows our geospatial data analysis using 33 Sentinel-2 satellite imagery within the state of Missouri for the year 2022. Figure 2A on the left of the figure illustrates a map of Missouri, demarcated by latitude and longitude, overlaid with 33 distinct green rectangles signifying the spatial coverage of Sentinel-2 images. These images encapsulate notable geographic demarcations including county boundaries and principal urban areas. The central portion of the figure describes the process undertaken to standardize the raw satellite data, which entails normalizing the images to ensure homogeneity in terms of color and scale across all datasets. After this normalization, the images are methodically cropped and the size of each image chip is equal to 224 pixels by 224 pixels or 2.24 km by 2.24 km. These chips are superimposed on blue outlines that delineate 1052 census tracts of our study, suggesting a systematic intersection of medium-resolution satellite data with granular demographic units.

Figure 2B on the right provides a detailed view of the Sentinel-2 image coverage as it intersects with the state-designated census tracts. It presents the state of Missouri as a green plane, superimposed with an array of green squares, each representing the 82,500 image chips generated post-normalization. The distribution of these image chips across the state varies, with a higher density in certain regions, potentially reflecting areas of research study. This meticulous preprocessing routine is essential to prepare the data for the subsequent analytical procedures, ensuring precision and reliability in the results of the spatial analysis conducted within this research study.

### 3.4. Regression Analysis of Predictive Capability

We used the 10-fold cross-validation which is a widely used method for evaluating the performance of a model. It involves dividing the dataset into ten equal parts, or “folds”. The model is trained on nine folds and tested on the remaining one. This process is repeated ten times, with each fold serving as the test set once. The results from all ten iterations are averaged to provide an overall assessment of the model’s performance. This approach helps to minimize bias and variance in the evaluation, ensuring a more reliable and generalizable estimate of the model’s effectiveness [67,68,69].

In the evaluation of the 10-fold cross-validation fit we utilized three key metrics: Mean Squared Error (MSE), $R^{2}$, and Adjusted $R^{2}$. MSE is a measure of the average squared difference between the actual observed outcomes and the outcomes predicted by the model, providing a clear quantification of the model’s prediction error [70]. Lower MSE values indicate better model accuracy. $R^{2}$, also known as the coefficient of determination, assesses the proportion of the variance in the dependent variable that is predictable from the independent variables [71]. It offers an insight into the goodness of fit of the model, with values closer to 1.0 suggesting a better fit [71]. Adjusted $R^{2}$ adjusts for the number of predictors in the model, providing a more accurate measure of the goodness of fit, especially important in models with a high number of predictors [72]. These metrics collectively offer a comprehensive evaluation of 10-folds cross-validation model performance.

## 4. Results

Figure 3 shows a choropleth map of obesity prevalence among the various census tracts within the state of Missouri for the year 2022. In the map, lighter shades of red are lower obesity rates, and darker hues correspond to higher obesity rates. The scale itself delineates a range starting at 25%, represented by a light red, and progresses to 50%, indicated by a dark red color. Notably, the map reveals a significant variation in obesity rates across different regions, with some census tracts exhibiting markedly higher rates and others reflect lower obesity rates.

Table 1 presents a detailed summary of key metrics across various regions such as population, area (in square Km), obesity crude prevalence (%), and number of chips. The data in the table includes the minimum, median, and maximum values for each metric, providing a clear understanding of the range and central tendencies within the dataset. For population, the minimum value recorded is 102, the median is 4058, and the maximum reaches 75,569, reflecting the diversity in population sizes across different areas. The area of the regions varies significantly, with the smallest being just 0.49 square Km, the median at 20.19 square Km, and the largest extending to 1787.47 square Km. In terms of obesity rates, the lowest rate observed is 23%, the median stands at 39.20%, and the highest rate is 53.7%, indicating varied health metrics across the regions. Finally, the number of chips ranges from a minimum of 1 to a maximum of 442, with a median of 14, highlighting different levels of chip distribution or consumption.

In Figure 4, highlights the spatial relation between our 63,592 image chips and the 1052 census tracts within the state of Missouri. Figure 4A provides an overview of the state of Missouri along with two examples of the chip and census tract spatial relationship complexity, marked by a blue boundary indicative of regional limits such as census tracts boundary. The green grid lines (squares/tiles) represent the image chips, which serve as discrete units of spatial data collection or observation points. Figure 4B Zoomed-in view around the Boone county area, narrows the focus to a smaller region, providing a more detailed look at the grid alignment of the image chips with respect to Boone County’s geography. A red square highlights a specific area within Boone County, presumably to denote an area of special interest or higher detail study. This zoomed-in perspective allows for an appreciation of the granularity of the image chip distribution within the context of finer geographical boundaries.

Figure 4C shows individual image chip that intersects seven distinct census tracts. This visualization captures the specific overlap between the selected image chip and seven distinct census tracts, allowing for a focused examination of these interactions. This image chip intersects with one of the highest number of census tracts (0002, 0003, 0005, 0009, 0010, 0021 and 0022) and it is located in Boone County. Figure 4D is a single census tract, specifically “Census Tract 0608”, with 150 intersecting image chips. This census tract is among the areas with one of the highest number of image chips. This level of granularity reveals a concentrated cluster of data points, potentially signifying a region of particular interest or higher measurement intensity.

Overall, the progression from a broad overview to a detailed view in the four figures systematically demonstrates the methodological approach of correlating medium-resolution satellite imagery with detailed census tract data. Such a multi-scale visual representation aids in understanding the spatial analysis techniques used in our research study, highlighting how local demographic characteristics may be inferred or validated through the meticulous overlay of image chips onto census tract maps. This meticulous approach underscores our research’s emphasis on precision and spatial specificity in its analysis.

Figure 5 presents a series of scatter plots, each corresponding to one of ten folds from a 10-fold cross-validation procedure, used to assess the performance of a predictive model for obesity rates. Cross-validation is a machine learning method used to evaluate predictive models by partitioning the original sample into a training set to train the model, and a test set to evaluate it.

In every individual plot, the horizontal axis represents the actual obesity rates (%) and the vertical axis represents the obesity rates as predicted by the model (%). Each plot is labeled as “Fold 1” through “Fold 10”, indicating the sequential partitioning of the dataset used for validation. In all plots, the red dashed line represents the theoretical perfect prediction.

A visual inspection across all ten folds shows that the model’s predictions are reasonably well aligned with the actual data, as indicated by the cluster of dots around the linear trend line. However, there is variability among the folds, with some showing tighter clustering around the line (indicating more accurate predictions) and others showing more spread (indicating less accurate predictions). This is expected in cross-validation due to the random partitioning of data into different folds, and it allows for the assessment of the model’s robustness and generalizability.

The uniformity in the distribution of points across the ten folds indicates that the model is consistently making predictions with a similar degree of accuracy. There is no single fold that appears to be an outlier in terms of prediction quality, which suggests that the model is stable and performs equally well across different subsets of the data.

### 4.1. Generalized Linear Model Regressor (GLM)

Figure 6 shows the actual obesity rates against those predicted by the GLM. The horizontal axis denotes the actual obesity rates derived from empirical observations, while the vertical axis corresponds to the predicted obesity rates yielded by the machine learning model. Each point on the plot represents a single census tract, with its position reflecting the actual rate on the x-axis against the model’s prediction on the y-axis. The red dashed line runs through the data points, indicating the linear regression model’s line of best fit. This line encapsulates the general direction that the model attributes to the relationship between the actual and predicted values of obesity rates. In an ideal scenario where predictions perfectly match the actual rates, all data points would align precisely along this line.

In the process of evaluating the predictive performance of GLM for obesity rates, the dataset was partitioned into two subsets; 80% of the data was used for training the model, ensuring it could learn the underlying patterns and relationships effectively. The remaining 20% (selected at random) constituted the testing set, which provided a separate and unbiased evaluation of the model’s predictive accuracy. This approach is critical for validating the model’s ability to generalize to new, unseen data and for minimizing the risk of overfitting. Figure 6 is annotated with key statistical metrics that quantify the model’s predictive performance. The model is able to moderately predict the obesity rate from the satellite imagery with a mean squared error (MSE) of 18.64, an $R^{2}$ equal to 0.44, and an adjusted $R^{2}$ value equal to 0.43. The plot and the metrics together provide a concise summary of the model’s performance and highlight areas where further model refinement or additional data collection might be necessary to improve the accuracy of the obesity rate predictions.

### 4.2. Random Forest (RF)

Figure 7 presents a scatter plot visualizing the comparison between actual and predicted obesity rates using a Random Forest regression model. The horizontal axis represents the actual obesity rates expressed as a percentage, while the vertical axis represents the predicted obesity rates, also in percentage terms. Each blue dot on the plot represents a data point where the actual and predicted obesity rates for a given observation are plotted against each other.

The red dashed line represents the line of best fit, illustrating the relationship between the actual and predicted values. Ideally, if the predictions were perfect, all dots would fall on this line, which would indicate a 1:1 correspondence between actual and predicted rates. However, the spread of the dots around this line suggests variability in the model’s accuracy, with some predictions being more accurate than others.

In our study, we utilized a Random Forest algorithm to forecast obesity rates across various census tracts. To ensure a robust assessment of the model’s predictive power, we divided our dataset into two distinct portions; 80% was allocated for training the model, allowing it to learn from a substantial majority of the data, while the remaining 20% (selected at random) was set aside for testing purposes. This testing set served as a crucial benchmark for evaluating the model’s efficacy in accurately predicting obesity rates on new, unseen data, thereby safeguarding against overfitting.

The inset box at the bottom of the plot provides key statistical metrics evaluating the performance of the model. The Mean Squared Error (MSE) is reported as 17.35, which gives the average of the squares of the errors that is, the average squared difference between the estimated values and the actual value. The $R^{2}$ value is 0.48, suggesting that approximately 48% of the variability in the actual obesity rates can be explained by the model. The Adjusted $R^{2}$ is 0.47, slightly lower than the $R^{2}$, which takes into account the number of predictors in the model and provides a more adjusted estimation of the goodness of fit.

The positioning of the dots in relation to the line of best fit and the reported $R^{2}$ values suggest that while the model has a moderate predictive power, there is a significant portion of the variance in obesity rates that is not captured by the model. This could be due to the complexity of factors affecting obesity rates that are not included in the model or due to the inherent limitations of the Random Forest algorithm when applied to this particular dataset. As we mentioned previously for GLM, the RF plot and metrics together also provide a concise summary of the model’s performance and highlight areas where further model refinement or additional data collection might be necessary to improve the accuracy of the obesity rate predictions.

### 4.3. Features Importance

Table 2 provides a breakdown of the top ten features in terms of their importance percentages, derived from the 10-fold-cross validation model aiming to predict the obesity rates using Deep Neural Visual Features (DNVFs). Feature 1112th is the most significant, holding an importance of 15.01%, indicating its strong influence on the model’s predictions regarding obesity rates. The subsequent features, although less important than Feature 1112th, still contribute notably to the model’s prediction and include Features 0095th and 1314th, with importance percentages of 3.23% and 2.47%, respectively.

The coefficients in the third column indicate the strength and direction of the relationship between the identified features and obesity prevalence. Positive values suggest a direct correlation, where higher values of the feature are associated with higher obesity rates; whereas negative values indicate an inverse relationship, suggesting that higher feature values are associated with lower obesity rates. For example, Feature 1112th shows a strong negative correlation of −0.6061, implying that higher values of this feature are associated with lower obesity rates.

In Figure 8, Feature 1112th is identified as the most significant predictor of obesity rates among the 2048 features, marking its top rank in importance. A comparative analysis of the heat maps for Feature 1112th and obesity rates reveals a strong correlation between the two, as evidenced by Table 2 data where Feature 1112th has an importance of 15.01% and a predicted obesity correlation coefficient of −0.6061. This negative correlation suggests that higher values of Feature 1112th are associated with lower obesity rates. Feature 1112th exhibits high values in densely populated urban areas and frequently visited tourist destinations. The cities of Kansas City (population: 510,704), St. Louis (population: 281,754), Springfield (population: 170,188), and Columbia (population: 129,330) are the top four most populated cities in Missouri, according to the 2023 Population Estimates Program and the 2022 American Community Survey. All these cities are prominently highlighted on the heat map for Feature 1112th, aligning with their lower obesity rates due to the negative correlation indicated in the Table 2. Interestingly, the area around Branson, Missouri, despite having a relatively low full-time resident population of 12,897, also shows high values for Feature 1112th. Branson’s high feature values can be attributed to its status as a major tourist destination, which attracts millions of visitors annually. This suggests that Feature 1112th captures geospatial aspects related to urban development and human activity, such as:High Building Density and Commercial Structures: Frequent clustering of buildings and commercial areas.Extensive Transportation Networks: Well-developed roads, highways, and parking facilities.Presence of Recreational Areas: Parks, theaters, and other recreational facilities indicating significant human activity.Impervious Surfaces: High proportion of concrete and asphalt surfaces.Shadow Patterns: Distinct patterns cast by tall structures.

The correlation between the heat maps of Feature 1112th and obesity rates, along with the feature’s association with densely populated and highly visited areas, indicates that Feature 1112th likely reflects critical geospatial aspects of urban environments. These aspects include building density, transportation networks, commercial and recreational infrastructure, and other indicators of human activity and development. Further research is necessary to decompose this feature and understand the specific attributes it encompasses, which could inform targeted public health interventions and urban planning policies.

### 4.4. Additional Top-5 Features

Figure 9, illustrates the geospatial distribution of two significant features across Missouri, identified as Feature 0095th and Feature 1314th, which are ranked as the second and third most important features relative to obesity rates. It is noteworthy that higher values of these features, particularly concentrated in urban centers such as Kansas City and St. Louis, are associated with lower obesity rates. This suggests that these features, which could represent factors such as access to recreational facilities or health services, play a crucial role in mitigating obesity rates in these areas.

Figure 10, presents two heat maps illustrating the spatial distribution of important features across Missouri, identified as Feature 0767th and Feature 0239th. Higher values of these features are associated with lower obesity rates across Missouri’s various regions. The visual patterns suggest areas where targeted interventions might be beneficial in reducing obesity prevalence.

Table 3 provides a numerical summary of the performance metrics for a 10-fold cross-validation of a regression model predicting obesity rates. Each row in the table corresponds to one of the ten folds used in the cross-validation process. The columns present the MSE, $R^{2}$, and Adjusted $R^{2}$ for each fold. The MSE is a measure of the average squared difference between the observed actual outcomes and the outcomes predicted by the model. The $R^{2}$ value indicates the proportion of the variance for the dependent variable that’s explained by the independent variables in the model, while the adjusted $R^{2}$ accounts for the number of predictors in the model and provides a more adjusted measure of the goodness of fit.

### 4.5. DCNN Model Explainability

Figure 11 showcases four distinct heatmaps analyzing obesity rates and prediction accuracy across the state of Missouri. Figure 11A is a heatmap which displays the actual obesity rates, where color intensities represent the percentage of the population affected by obesity, ranging from 25% to 55%. Darker shades correspond to areas with higher obesity rates, allowing for the identification of regions where obesity is more prevalent.

Figure 11B illustrates the obesity rates as predicted by a statistical model. The visual similarity between Figure 11A,B suggests a close alignment between predicted and actual rates, indicating the model’s effectiveness in mirroring the geographical distribution of obesity.

Figure 11C applies a filter to the absolute RMSE values, excluding any areas where the RMSE is lower than 4. This effectively removes more accurately predicted regions, highlighting only those areas where the predictive model’s errors are above this threshold. The resulting map is a patchwork of colors where only the highest errors in prediction are shown, offering a focused view on the model’s limitations.

Figure 11D refines the approach in Figure 11C by adjusting the RMSE filter to include areas with RMSE values of 2.5 and above. The same color ratio is maintained to ensure consistency in visual interpretation across Figure 11C,D. As a result, this figure shows a broader range of prediction errors compared to Figure 11C, providing a more nuanced understanding of the model’s performance across the region.

The progression from Figure 11A,B to Figure 11C,D demonstrates a methodical approach to analyzing model accuracy. The initial heatmaps set the stage by presenting actual and predicted rates, while the latter heatmaps critically evaluate the model by highlighting regions with the most significant prediction errors. This step-by-step visualization emphasizes areas where the predictive model could be improved, guiding researchers and policymakers in directing their efforts to refine predictive analytics and address obesity more effectively in regions with higher prediction errors.

Figure 11E features a curve that represents the signed error between predicted and actual values across different census tracts. The horizontal axis is labeled “Signed Error (%)” and indicates the percentage error of prediction, with negative values on the left side representing under-predictions and positive values on the right side indicating over-predictions. The vertical axis is labeled “Census Tract Ranked by Error”, suggesting that each census tract has been ranked based on the magnitude of the prediction error.

The curve starts on the left with a steep incline in the red zone, indicating a significant under-prediction of obesity rates in some census tracts (those with a high negative signed error). As the curve moves to the right, the color transitions to green, and the slope decreases until the error approaches zero. The zero point on the horizontal axis likely represents the point at which the prediction exactly matches the actual obesity rate.

Beyond the zero point, the curve extends into the “Over Predict” zone, where the signed error becomes positive, indicating census tracts where the model has overestimated the obesity rate. The curve in this region is relatively flat and continues in green, suggesting fewer tracts with over-prediction compared to under-prediction.

The shape of the curve indicates that the distribution of errors is skewed; there are more census tracts with substantial under-predictions than over-predictions. This could imply that the model used for prediction has a systematic bias or that certain factors leading to higher obesity rates are not being adequately captured in those tracts with higher negative errors.

Table 4 is the data extracted from the larger dataset, and is related to our study on obesity prevalence within various census tracts. Each row corresponds to a specific census tract as indicated by the “GEOID” column, which contains unique numerical identifiers. The “County” column lists the location of each census tract, with multiple entries for the urban counties of St. Louis and Jackson, and only one entry for the rural counties of Stoddard and Pemiscot.

The “Population” column lists the number of people residing in each census tract. These populations range from as low as 1651 to as high as 6669, suggesting a diverse set of tracts in terms of population size.

In the “Actual Obesity (%)” column, we see the actual percentage of the population within each tract that has been classified as obese. These percentages range from 26.90% to 52.80%, indicating a significant variation in obesity rates across different areas.

The “Predicted Obesity (%)” column shows the predicted obesity rates as for the same census tracts. The predicted percentages exhibit a similar range to the actual percentages but do not always align exactly, which is to be expected in predictive modeling.

The final column, “Signed Error (%)”, represents the difference between the predicted and actual obesity rates, expressed as a percentage. A positive signed error indicates that the model has over-predicted the obesity rate, while a negative signed error suggests under-prediction. For instance, the first row shows an over-prediction of 8.37% for the census tract with GEOID 29510124600, meaning the model estimated the obesity rate to be 8.37 percentage points lower than the actual rate. Conversely, the second row, with GEOID 29207470600, shows an under-prediction of 8.52%, where the model’s estimate was higher than the actual rate.

## 5. Discussion

The visual contrasts in Figure 3 underscore the presence of geographical health disparities that may warrant targeted public health interventions and resource allocation. However, the map also highlights the challenge of representing densely populated areas where census tracts are smaller, and thus, the corresponding data may be less discernible. In all, the study shows that in both over-sampled (large) census tracts and under-sampled (very small) census tracts the model performs well.

The MSE for each fold is a measure of the average squared difference between the observed actual outcomes and the outcomes predicted by the model. The values range from a low of 10.22 in Fold 6 to a high of 22.06 in Fold 2, indicating variability in the model’s predictive accuracy across different subsets of data [73,74,75]. This is a common phenomena referred to as finite sample effects [76], which indicates that with limited data, the specific examples in each fold can have a disproportionate influence on model training. Some folds might, by chance, contain more challenging or unusual examples.

The $R^{2}$ values range from as low as 0.011 in Fold 2 to as high as 0.661 in Fold 6. The $R^{2}$ metric provides a measure of how well the variation in the dependent variable (obesity rates) is explained by the model. In this context, Fold 6’s model explains 66.1% of the variance, which is quite high, whereas the model in Fold 2 explains only 1.1%, which is extremely low.

Adjusted $R^{2}$ is a modified version of $R^{2}$ [77] that adjusts for the number of predictors in the model. It is generally lower than $R^{2}$ as it penalizes excessive use of predictors. The Adjusted $R^{2}$ values closely follow the $R^{2}$ values, with the highest being 0.657 in Fold 6 and the lowest being 0.001 in Fold 2.

The last row provides the mean of the MSE, $R^{2}$, and Adjusted $R^{2}$ across all folds. The mean MSE of 14.69 suggests that on average, the model has a moderate level of prediction error. The mean $R^{2}$ of 0.477 indicates that, on average, the model explains about 47.7% of the variability in the actual data, which suggests a moderate fit. Similarly, the mean Adjusted $R^{2}$ of 0.471 is in line with the mean $R^{2}$, indicating a reasonable fit after adjusting for the number of predictors.

Table 3 highlights that the performance of the model varies significantly across the different folds, as evidenced by the range of values in MSE and $R^{2}$ metrics. This variation can arise due to the inherent differences in the data subsets or may indicate an underlying issue with the model’s stability. Table 3 underscores the importance of cross-validation to understand how the model might perform in general, rather than relying on the metrics from a single partition of the data. Overall, Table 3 provides a detailed breakdown of the model’s performance across different segments of the data, highlighting variations in model accuracy and fit.

Figure 11E provides a visual representation of the prediction errors across census tracts, with the curve’s profile offering insight into the model’s predictive performance and potential biases. The clear demarcation of under-prediction and over-prediction zones also helps in identifying the direction of the errors for subsequent analysis and model refinement.

Table 4 is likely used to analyze the accuracy of the obesity rate predictions by the model, providing insight into where the model tends to overestimate or underestimate obesity prevalence. This information could be valuable for refining the model or for directing public health resources to areas where obesity is more prevalent than initially predicted.

Despite the promising results, our study has some limitations. First, the estimates of obesity prevalence from the Behavioral Risk Factor Surveillance System rely on self-reported measurements of height and weight, which are subject to bias and often result in an underestimation of the true rate of obesity [37,39]. Variations in the timing between when the obesity data and the satellite images are collected can also introduce biases into our analysis. One of the primary limitations of our study pertains to the dataset’s size and geographical coverage. The research was confined to 1052 census tracts within the state of Missouri, limiting the generalizability of the findings.

Although these tracts were selected to represent a diverse range of urban and rural areas, they do not encompass the neighboring states and their varied demographic and geographic profiles. Furthermore, the limited number of census tracts might not provide a sufficiently robust dataset for more complex machine learning models [78]. The pre-trained ResNet-50 [59] is not fully optimized for the specific nuances of satellite image analysis related to obesity rate prediction. Our study’s findings must therefore be interpreted with caution, acknowledging that the employed models, although advanced, might not capture the complete range of factors influencing obesity rates as discernible from satellite imagery. Additionally, training the model to adjust its weights to satellite data and the chosen census tracts should yield significantly better results.

## 6. Conclusions

In summary, the main findings of our research study are to emphasize the novel methodology using DNVF to analyze and predict the obesity prevalence for different census tracts in Missouri. In conclusion, the methodology employed in this research holds great promise for enhancing our understanding and predicting obesity. It has the potential to inform more effective community-level and policy-driven strategies to combat this complex health issue.

Although the study is innovative in its approach. The reliance on self-reported data could introduce biases, and the focus solely on Missouri limits the generalizability of the findings. These aspects, however, present valuable opportunities for future research. In the future, it will be important to expand the geographical scope of this study to include diverse regions, enhancing the generalizability of our findings. We also plan to integrate additional data sources, like socio-economic and health-related factors, to provide a more comprehensive analysis. The refinement of machine learning techniques, particularly advanced models and deep learning, will be pivotal in improving prediction accuracy. Longitudinal and geospatial studies are envisioned to observe temporal changes in obesity rates and other population health diseases.

## Figures and Tables

**Figure 1 ijerph-21-01534-f001:**
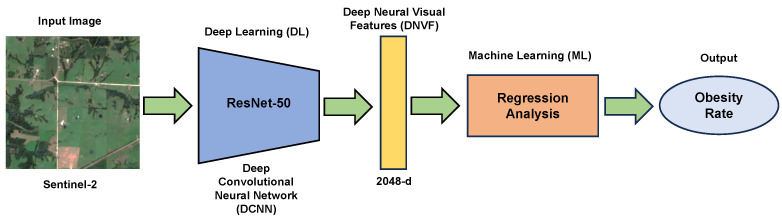
Flowchart illustrating the estimation of obesity rates from satellite imagery using a combination of deep learning with ResNet-50 architecture and machine learning regression analysis.

**Figure 2 ijerph-21-01534-f002:**
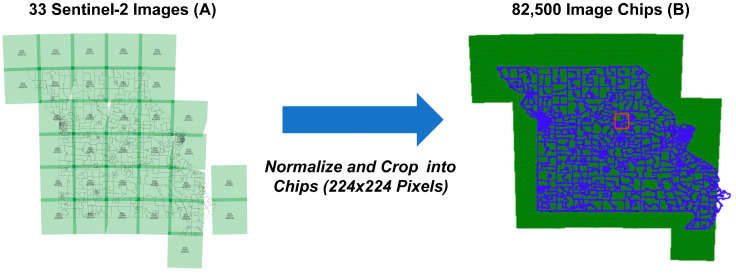
Data processing workflow for Sentinel-2 satellite imagery within Missouri in 2022. (**A**) Displays the geographic coverage of 33 Sentinel-2 images across Missouri, with county boundaries. The central diagram outlines the normalization process and the cropping of images into 224 × 224 pixel chips. (**B**) Illustrates the distribution of 82,500 resultant image chips. The red box represents Mid-Missouri area and Boone County.

**Figure 3 ijerph-21-01534-f003:**
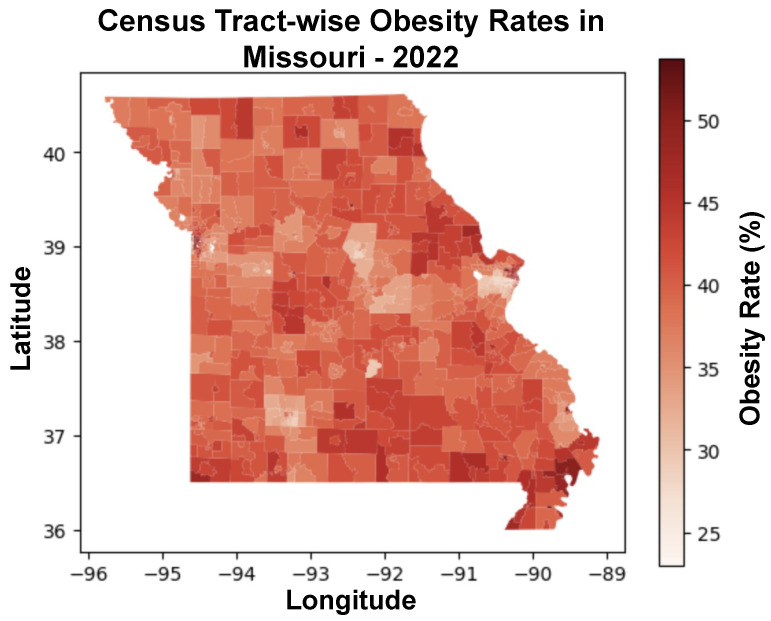
Choropleth map displaying the distribution of obesity rates percentage for individuals across Missouri census tracts in 2022. The variations in the color intensity reflect the range of obesity prevalence, with darker red indicating higher obesity rates. The color scale to the right quantifies the obesity rates corresponding to each color shade.

**Figure 4 ijerph-21-01534-f004:**
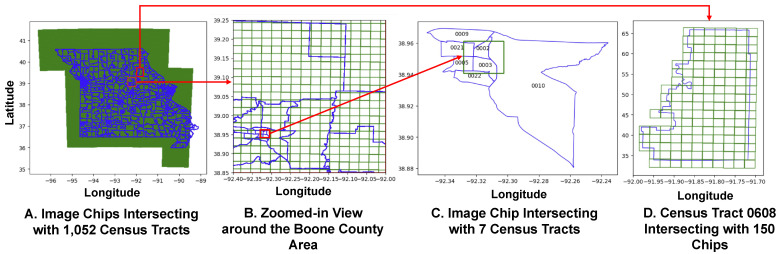
Multiscale analysis of satellite image chips and census tracts in Missouri. (**A**) exhibits a statewide view with image chips overlaying 1052 census tracts, indicating extensive data coverage. (**B**) zooms into the Boone County area, detailing the alignment of image chips to local geography. (**C**) details individual image chips boundaries, illustrating their overlap with seven distinct census tracts (numbered for reference). (**D**) further narrows down to Census Tract 0608, demonstrating the intersection with 150 specific image chips for granular analysis. The figure highlights the granularity and density of data distribution within the geographic study area.

**Figure 5 ijerph-21-01534-f005:**
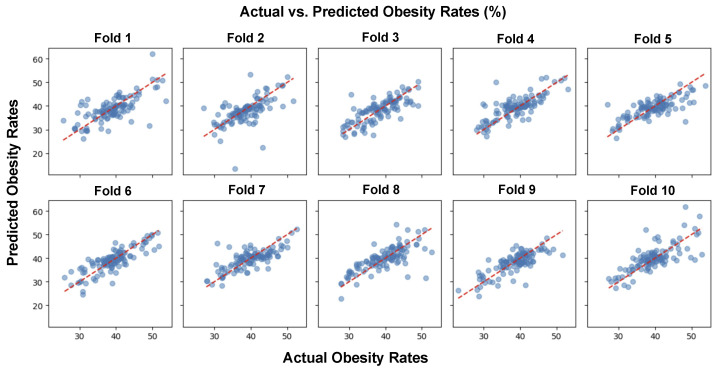
Scatter plots show the relationship between actual and predicted obesity rates using a GLM machine learning model across 10 distinct cross-validation folds. The red dashed line represents perfect prediction accuracy.

**Figure 6 ijerph-21-01534-f006:**
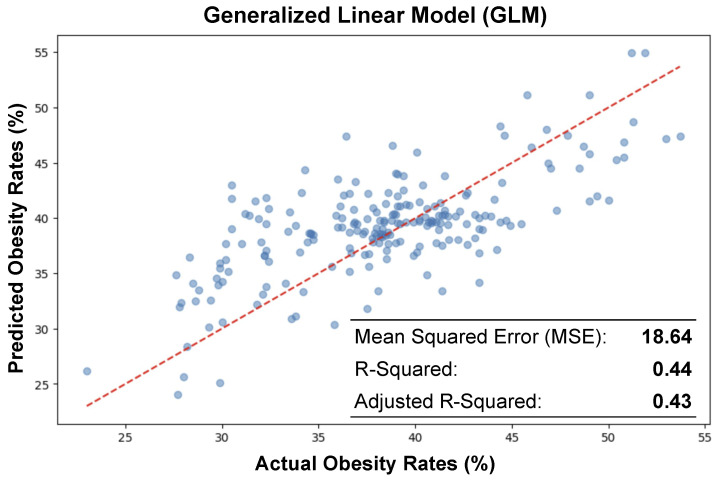
Scatter plot displaying the relationship between actual and predicted obesity rates using Generalized Linear Regression (GLM), illustrating a moderate degree of correlation with an $R^{2}$ value of 0.44 and an adjusted $R^{2}$ of 0.43. The close fit is further evidenced by a moderate Mean Squared Error (MSE) of 18.64. These metrics are provided to assess the accuracy of the model predictions.

**Figure 7 ijerph-21-01534-f007:**
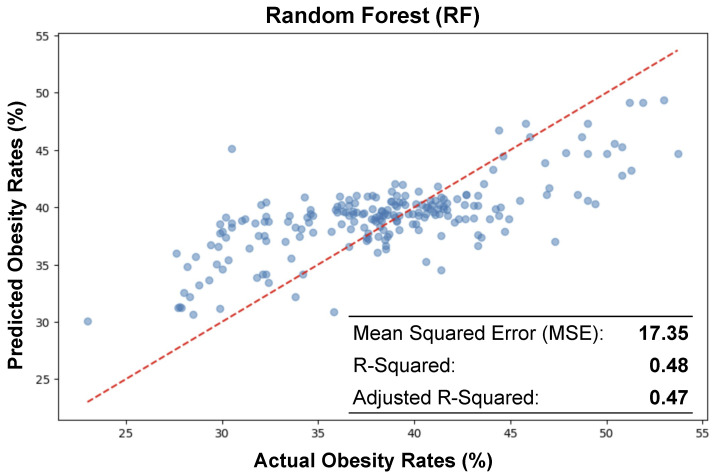
Scatter plot of the relationship between actual and predicted obesity rates using random forest, illustrating a moderate correlation with an $R^{2}$ value of 0.48 and an adjusted $R^{2}$ of 0.47. The close fit is further evidenced by a moderate Mean Squared Error (MSE) of 17.35.

**Figure 8 ijerph-21-01534-f008:**
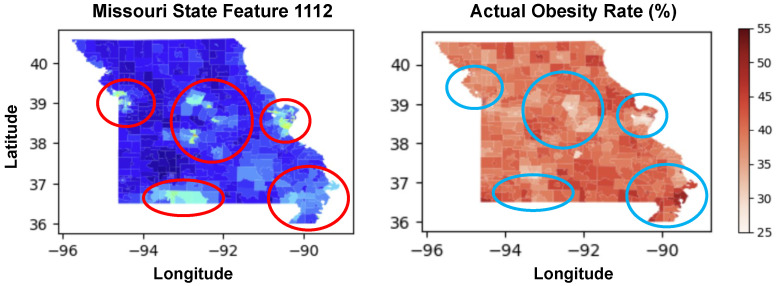
The left map shows spatial distribution of Feature 1112 across Missouri, with red circles highlighting areas of high values (urban areas). The right map depicts actual obesity rates (%) across the state, with blue circles indicating regions with lower obesity prevalence. Notable discrepancies between feature values and obesity rates can be observed in several regions.

**Figure 9 ijerph-21-01534-f009:**
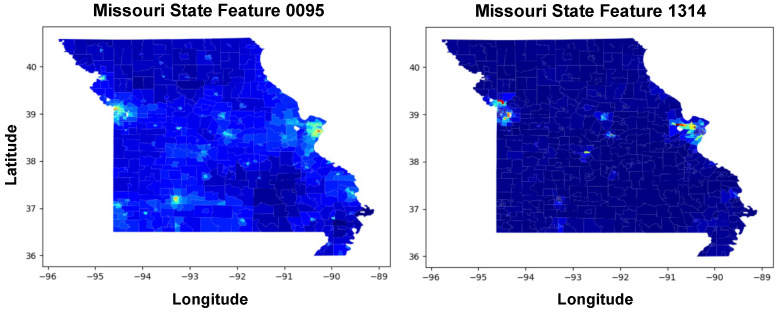
The spatial distribution of the 2nd and 3rd most important features across Missouri, (**left**) Feature 0095th and (**right**) Feature 1314. Urban areas, particularly around Kansas City for Feature 0095 and St. Louis for Feature 1314, show significant concentrations of these features.

**Figure 10 ijerph-21-01534-f010:**
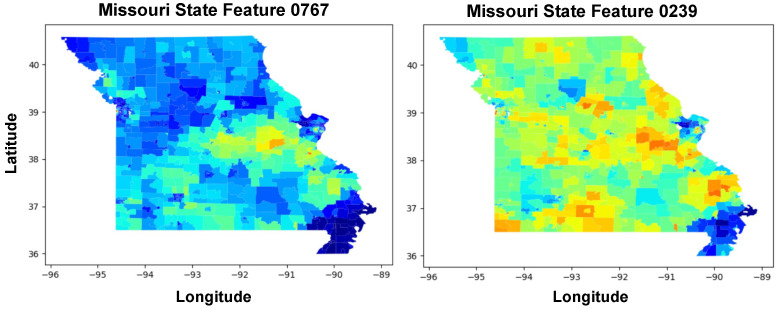
Spatial distribution of the 4th and 5th most important features across Missouri, (**left**) Feature 0767 and (**right**) Feature 0239.

**Figure 11 ijerph-21-01534-f011:**
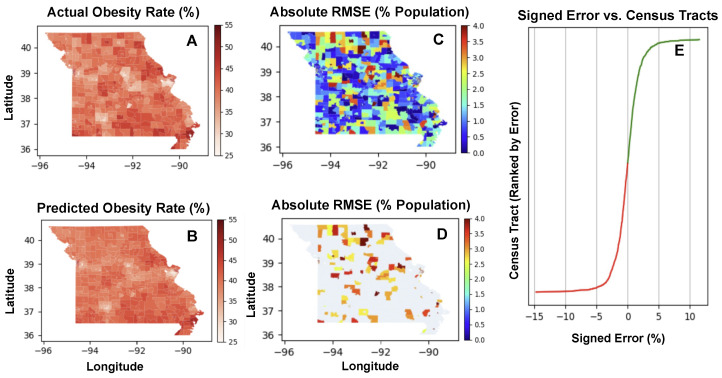
Geospatial analysis of obesity rates and prediction accuracy in Missouri. (**A**) displays the actual obesity rates, while (**B**) shows the predicted rates, both using a color gradient to represent percentages. (**C**) highlights areas with significant predictive errors by filtering out RMSE values below 4, focusing on regions where the model’s accuracy is lower. (**D**) refines this analysis by presenting a broader error distribution, including RMSE values of 2.5 and above, using the same color gradient for consistency. (**E**) Curve illustrating the signed error distribution of predicted obesity rates across census tracts. Negative signed errors indicate underpredictions (shown in red) and positive errors indicate overpredictions (shown in green). The census tracts are ranked by the magnitude of error, highlighting the asymmetry in predictive accuracy and potential systematic bias in the model.

**Table 1 ijerph-21-01534-t001:** A summary of key regional metrics, covering total population, area (square Km), obesity crude prevalence, and number of chips intersected with each tract.

	Population	Area (Sqr. Km)	Obesity Prevalence (%)	Number of Chips
Min	102	0.49	23.00	1
Median	4058	20.19	39.20	14
Max	75,569	1787.47	53.70	442

**Table 2 ijerph-21-01534-t002:** Lists the top ten visual features ranked by their importance in our obesity rate prediction model. The percentages indicate the relative importance of each feature in influencing the model’s predictions, with higher values signifying greater influence. The values of the correlation coefficient, highlighting the magnitude of each feature’s impact.

Feature #	Importance (%)	Correlation Coefficient
1112	15.01	−0.6061
0095	3.23	−0.2196
1314	2.47	−0.2079
0767	2.07	0.1542
0239	2.05	−0.1526
1253	1.05	0.1140
0895	1.04	0.0932
1126	0.87	−0.0536
0338	0.84	−0.0434
0668	0.75	−0.0061

**Table 3 ijerph-21-01534-t003:** Provides a numerical summary of the performance metrics for a 10-fold cross-validation of a regression model predicting obesity rates.

Fold #	MSE	$R^{2}$	Adjusted $R^{2}$
1	18.54	0.406	0.400
2	22.06	0.011	0.001
3	12.56	0.538	0.534
4	13.07	0.546	0.541
5	13.58	0.559	0.555
6	10.22	0.661	0.657
7	12.81	0.537	0.533
8	16.68	0.485	0.480
9	11.06	0.567	0.563
10	16.36	0.456	0.451
Mean	14.69	0.477	0.471

**Table 4 ijerph-21-01534-t004:** The table lists the GEOID, County, Population, and actual versus predicted obesity percentages for selected census tracts, along with the signed error indicating the discrepancy between predicted and observed values.

GEOID	County	Population	Actual Obesity (%)	Predicted Obesity (%)	Signed Error (%)
29510124600	St. Louis	1845	46.00	37.63	8.37
29207470600	Stoddard	4968	39.80	48.32	−8.52
29095005100	Jackson	1651	34.00	42.76	−8.76
29095016100	Jackson	2046	49.80	40.42	9.38
29095008200	Jackson	2765	30.50	40.14	−9.64
29510119300	St. Louis	5454	26.90	37.82	−10.92
29095015400	Jackson	3826	52.80	41.61	11.19
29155470200	Pemiscot	3613	50.80	39.31	11.49
29095006100	Jackson	2542	48.30	61.83	−13.53
29189213400	St. Louis	6669	37.40	52.07	−14.67

## Data Availability

Data from this study will be made available upon request.

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
