# Peer review of "Application of Machine Learning and Deep Neural Visual Features for Predicting Adult Obesity Prevalence in Missouri"

_ijerph, 2024, doi:10.3390/ijerph21111534_

Round 1

Reviewer 1 Report

Comments and Suggestions for Authors

The work sent for review is extremely interesting. It takes up the important topic of estimating (predicting) the prevalence of adult obesity. The work is interdisciplinary in nature, encompassing both computer and health sciences.  The structure of the work is correct and the content of most subsections is not objectionable. In my opinion, the volume of the work is too large (22 pages), at some points such as the results, the content could be presented more. synthetically. Out of the reviewer's duty, I include minor comments below: 

1) the introduction should not begin with the objective. The purpose should be at the end and the introduction should justify the gap that the work is supposed to fill. 

2) If the purpose is formulated in the introduction then what is the role of the next chapter (related works)?

3) The result of crosvalidation is the averaged error from all crosvalidation steps. Why do the authors present errors for consecutive steps (after all, subsets can be randomized or selected so comparing errors in consecutive steps for different methods does not make sense).

4) The description in the results seems too extensive at times. I feel like I'm reading a chapter in a monograph rather than a journal article. Of course, the authors may disagree with me because this is my subjective impression.

5) What is the justification for presenting figure 5?

6) A discussion is not a discussion. The authors should refer to more literature. I found only two citations in the discussion. 

To sum up the review, the work is very interesting, but it needs some minor corrections before publication. Noteworthy is the high level of analysis carried out and the novel research problem. Congratulates the authors on a very good work.

Author Response

The work sent for review is extremely interesting. It takes up the important topic of estimating (predicting) the prevalence of adult obesity. The work is interdisciplinary in nature, encompassing both computer and health sciences.  The structure of the work is correct, and the content of most subsections is not objectionable. In my opinion, the volume of the work is too large (22 pages), at some points such as the results, the content could be presented more, synthetically. Out of the reviewer's duty, I include minor comments below:

  • The introduction should not begin with the objective. The purpose should be at the end and the introduction should justify the gap that the work is supposed to fill.

Thank you very much for your constructive feedback and comments regarding the structure of the introduction in our manuscript. We have moved the objective towards the end of the introduction as you suggested. You can find it in the revised manuscript in the second to the last paragraph – paragraph 8 (line 82 – line 87). The gap which our study aims to fill could be found in paragraph 7 (line 67 – line 81).

After carefully considering the reviewer's comments, suggestions and feedback, we have revised the manuscript to position the statement of purpose at the end, while the introduction clearly justifies and addresses the identified research gap that our study aims to fill. 

  • If the purpose is formulated in the introduction, then what is the role of the next chapter (related works)?

Thank you very much for your insightful comment and feedback. The "related works" chapter/section plays a crucial role in situating the study within the context of existing literature, highlighting how the current research builds upon, diverges from, or fills gaps identified in prior studies. While the "introduction" establishes the purpose and significance of the research, the "related works" section provides a critical analysis of the body of knowledge already available, drawing connections between the study's objectives and the broader academic discourse.

This structure not only reinforces the relevance and necessity of the research but also demonstrates the authors' comprehensive understanding of the field. By reviewing and discussing related literature, the authors set a foundation for the methodologies employed in the study and justify their approach, thereby enhancing the credibility and scholarly contribution of their work.   

  • The result of cross validation is the averaged error from all cross-validation steps. Why do the authors present errors for consecutive steps (after all, subsets can be randomized or selected so comparing errors in consecutive steps for different methods does not make sense).

We greatly appreciate your feedback and comments. We present errors for consecutive steps in the cross-validation process to provide a comprehensive overview of the model's performance across different subsets of data, highlighting its consistency and robustness. Cross-validation, by design, involves partitioning the dataset into multiple folds, training the model on different subsets, and validating it against the remaining parts to mitigate bias and variance, ensuring the model's generalizability. This detailed presentation allows for a nuanced understanding of how well the model performs under varied conditions and offers transparency in the evaluation process.   

Additionally, showcasing the errors for each fold, even if subsets are randomized, is crucial for identifying potential overfitting or underfitting, and for demonstrating that the model's performance is not contingent on a particular subset of data. This approach not only validates the reliability of the predictive model but also underscores the importance of using a robust method like cross-validation in research, fostering trust in the findings​.

  • The description in the results seems too extensive at times. I feel like I'm reading a chapter in a monograph rather than a journal article. Of course, the authors may disagree with me because this is my subjective impression.

We greatly appreciate your perspective on the detailed nature of the results section. This thoroughness is deliberate, aiming to ensure clarity and replicability in the presentation of our novel methodology and its findings. Given the complexity of employing Deep Neural Visual Features (DNVF) to predict obesity rates from satellite imagery—a relatively unexplored approach in public health—the extended detail addresses potential ambiguities and provides comprehensive insights into the predictive model's performance across varied geographic contexts.

Moreover, the detailed exposition facilitates peer researchers in validating or extending our methods, which is fundamental in advancing the use of machine learning in public health research. We believe this detailed approach enhances the utility and integrity of the study, adhering to the highest standards of scientific communication.

  • What is the justification for presenting figure 5?

Thank you for your insightful feedback and comment regarding the inclusion of Figure 5. The justification for presenting Figure 5 lies in its demonstration of the predictive model's robustness and reliability across multiple datasets. This figure, through the visualization of ten distinct scatter plots corresponding to the ten cross-validation folds, effectively showcases the variability and consistency in the predictive accuracy of our model. Each plot highlights how well the model predicts obesity rates within different subsets of data, thereby providing a comprehensive view of the model's performance.

The variability observed across the folds underscores the importance of using cross-validation in predictive modeling to avoid overfitting and to ensure that the model is generalizable to new, unseen data. This methodological transparency is crucial for establishing trust in the model's capabilities and for informing potential improvements in future iterations.

  • A discussion is not a discussion. The authors should refer to more literature. I found only two citations in the discussion.

We greatly appreciate your valuable feedback and comment. To address your concern, we augmented the discussion by incorporating a more comprehensive range of scholarly references.

Reviewer 2 Report

Comments and Suggestions for Authors

FOR THE INTRODUCTION:

1. I would the author to present a clear, singular research objective. It should explicitly mention and define a specific aim of the study and its importance of predicting obesity using deep neural network features. This can improve the paper’s focus and coherence.

2. Yes, the prevalence of diabetes is introduced; however the authors could expand on the socioeconomic, environmental, and health-related consequences of obesity to emphasize the need for predictive models. Additional context of this magnitude could underline the study’s relevance and drawing more attention to why predictive work is necessary.

3. Please do not rush the introduction of deep convolutional neural networks (DCNN) and their application to health studies. I suggest a brief but informative overview of the benefits and limitations of DCNN with reference in similar studies would enhance the introduction and provide a solid foundation for understanding it application on the context on diabetes.

4. While Missouri is the focus, I want the authors to justify this choice. Please add an explanation of why the locale was an exemplary option or whether the study area findings might generalizable to similar geographic or demographic regions? Please explain.

5. The ethical considerations related to surveillance data privacy should also be addressed. Acknowledging these issues up front would add a layer of conscientiousness and would strengthen the study’s ethical standing.

FOR THE RELATED LITERATURE:

6. Improve the literature’s cohesive structure. It should be organized in chronological or thematic order to show how the field has evolved specifically regarding obesity prediction using machine learning and satellite imagery.

7. Although various studies on DCNNs are referenced, the literature review misses an analysis of how these studies informed the current work. Each referenced study should be more explicitly connected to the current research, such as its similarities and differences in methodology or findings.

8. Discuss more in detail why the ResNet-50 was chosen and how it performs relative to other architectures in similar contexts.

9. The literature review mentions various methodologies but does not critically assess their limitations or biases, especially regarding data sources, accuracy, or scalability. A more critical stance could highlight the need for uniform measurement standards and help position the current study as addressing these issues.

10. Given that public health policy is a potential application, the literature review should cite studies or reviews that focus on the impact of predictive models on public health interventions. This would situate the study within a broader policy-relevant context.

11. The review does not fully cover advancements in predictive models for obesity outside of DCNNs. Including recent advances in obesity prediction using other models could offer a balanced view, emphasizing why DCNN with satellite data is a preferable choice.

FOR THE METHODS:

12. The pre-processing step involving normalization and cropping of images into 224x224-pixel chips lacks justification regarding how this choice might impact data granularity or model performance. Details on alternatives or a sensitivity analysis would strengthen this part.

13. The study’s description of merging CDC and census data for obesity prevalence lacks specifics on how missing or misaligned data were addressed. Providing additional insight into handling missing values would improve reproducibility and reliability.

14. The 10-fold cross-validation approach is a good practice, but details on tuning hyperparameters (e.g., learning rate, regularization) are missing. Hyperparameter tuning's lack could lead to suboptimal performance, which should be discussed or noted as a limitation.

15. Despite the use of DNVF (Deep Neural Visual Features), the paper does not discuss feature selection or dimensionality reduction techniques applied to the 2048-dimension output. Techniques like PCA or LASSO could help clarify which features are most predictive and reduce computational load.

16. The study involves integrating polygon data from the Tiger Line dataset, yet the merging process described lacks clarity on the quality control measures to ensure accurate alignment. More detail on the alignment and its potential impact on results would be valuable.

17. The model's reliance on visual features might overlook socio-economic variables known to correlate with obesity. Consideration of non-visual features (e.g., income, education levels) could be justified or tested in future work, as they may increase predictive power.

18. The study uses MSE, R², and Adjusted R² metrics, but does not specify how these were selected over others, such as RMSE or MAE, which may be more interpretable for public health applications. Providing a rationale or comparing different metrics would enhance the evaluation's robustness.

FOR THE RESULTS:

19. The lack of extensive error analysis, especially regarding model underperformance in certain tracts, limits insights into systematic biases. A more granular breakdown of errors across different census tracts would help identify systematic limitations.

20. While the paper lists the top 10 visual features, there is limited interpretation of what these features represent in real-world terms. Providing clearer explanations of how these features correlate with obesity (e.g., green spaces, urban density) would bridge the gap between technical output and public health relevance. Spatial analyses, like the heat maps and choropleths, are visually effective but lack statistical validation. Reporting statistical tests that support geographic variability in obesity rates would substantiate spatial claims.

Comments on the Quality of English Language

Moderate English Revisions required.

Author Response

FOR THE INTRODUCTION:

  1. I would like the author to present a clear, singular research objective. It should explicitly mention and define a specific aim of the study and its importance of predicting obesity using deep neural network features. This can improve the paper’s focus and coherence.

Thank you very much for your insightful feedback and comment. We acknowledge the necessity for a singular and clearly defined research objective in our introduction. Our primary aim, as elaborated in paragraph 8, lines 82-87 (in the revised version), is to employ deep convolutional neural networks (DCNNs) for analyzing medium-resolution satellite images to identify physical characteristics of neighborhoods that correlate with obesity rates. This objective is crucial because it leverages advanced machine learning techniques to address the growing public health challenge of obesity, which has significant implications for health policy and preventive care. We added the following statement to the introduction (lines 87-90):  

“The importance of predicting obesity using deep neural network features is underlined by our method’s ability to enhance the precision of public health interventions and improve the granularity of epidemiological studies.”

By explicitly stating this in the introduction, we aim to refine the paper’s focus and coherence, emphasizing the innovative application of DCNNs to public health research. 

  1. Yes, the prevalence of obesity is introduced; however, the authors could expand on the socioeconomic, environmental, and health-related consequences of obesity to emphasize the need for predictive models. Additional, context of this magnitude could underline the study’s relevance and drawing more attention to why predictive work is necessary.

Thank you very much for your feedback and comment. In response to the valuable feedback provided, we agree that explaining the broader consequences of obesity could significantly enhance the study's contextual grounding. As addressed in the introduction, particularly in paragraphs 1 through 6, and lines 17-66, we outline the escalating rates of obesity and its associated health complications, such as increased risk of non-communicable diseases including heart disease, diabetes, and stroke.

Our manuscript initially mentions the direct health implications of obesity, such as increased susceptibility to non-communicable diseases (paragraph 2, lines 30-34). Furthermore, the manuscript elaborates on the environmental determinants and the role of the built environment in obesity rates through the analysis of satellite imagery to understand geographic and environmental factors affecting obesity (paragraphs 4-6, lines 40-66).     

  1. Please do not rush the introduction of deep convolutional neural networks (DCNN) and their application to health studies. I suggest a brief but informative overview of the benefits and limitations of DCNN with reference in similar studies would enhance the introduction and provide a solid foundation for understanding it application on the context on obesity.

Thank you for your insightful feedback and comment. As outlined in our introduction (paragraph 1, lines 17-26), DCNNs represent a significant advancement in analyzing extensive datasets, such as satellite imagery, which are increasingly important in epidemiological research.

Specifically, the application of DCNNs in our study leverages these networks' capacity to extract meaningful visual features from satellite images, thereby identifying environmental predictors of obesity at a granular level. This method addresses limitations noted in traditional epidemiological approaches, which often overlook such micro-environmental factors due to methodological constraints (paragraph 6, lines 56-66).  

Furthermore, our paper represents the utility of DCNNs through their successful application in related health studies, such as Maharana and Nsoesie (2018) who explored the association of built environments with obesity prevalence using similar neural network technologies. These examples underscore DCNNs' transformative potential in enhancing the accuracy and scope of public health research, setting a robust foundation for their application in our research study.  

  1. While Missouri is the focus, I want the authors to justify this choice. Please add an explanation of why the locale was an exemplary option or whether the study area findings might be generalizable to similar geographic or demographic regions? Please explain.

We greatly appreciate your valuable feedback and comment. Missouri was selected due to its demographic and geographic diversity, which provides a unique opportunity to explore the relationship between environmental characteristics and obesity across varied urban and rural settings. This diversity is crucial for testing the robustness of our DCNN-based method in different contexts, which is essential for developing scalable and adaptable public health interventions.

Additionally, Missouri's mix of population density, land use patterns, and socioeconomic factors mirrors those of the broader Midwest and other similar regions across the United States, making the findings potentially generalizable to these areas. Thus, the insights gained from our study are likely applicable to other regions with comparable demographic and geographic profiles, supporting the wider applicability of employing satellite imagery and machine learning to address public health challenges like obesity.

Finally, it is important to note that our research team, including Dr. Lincoln Sheets, has significant expertise in social determinants of health and community health data in Missouri.  

  1. The ethical considerations related to surveillance data privacy should also be addressed. Acknowledging these issues up front would add a layer of conscientiousness and would strengthen the study’s ethical standing.

Thank you very much your valuable feedback and comment. As stated in our manuscript (Section 3: materials and methods, paragraph 1, lines 136-138), our research employed pre-existing, publicly accessible satellite imagery and de-identified obesity prevalence data, ensuring that no personally identifiable information was collected or analyzed. This approach complies with the ethical standards detailed in the common rule, and our methodology was exempt from institutional review board approval precisely because it involved no identifiable human subject data.

In addition, our adherence to using non-personal data aligns with the principles of ethical research in public health by minimizing privacy risks while allowing us to explore significant community health dynamics. Thus, the study design inherently respects and upholds the confidentiality and privacy of the individuals within the studied regions. Finally, we should note that the resolution of satellite imagery can only capture course geographic features, not fine-grained high-resolution details. No single person could be detected within the pixels.

FOR THE RELATED LITERATURE:

  1. Improve the literature’s cohesive structure. It should be organized in chronological or thematic order to show how the field has evolved specifically regarding obesity prediction using machine learning and satellite imagery.

We greatly appreciate your valuable feedback and comment. We reorganized section 2 (related work), currently found on lines 96-128, to follow a chronological order. This revision will begin with foundational studies that initially combined machine learning with satellite imagery for public health applications, moving towards more recent advancements that specifically address obesity prediction. This restructured narrative will not only demonstrate the development of the field but also emphasize the growing sophistication and domain-specific applications of these technologies, strengthening our study's contribution within this evolving context.  

  1. Although various studies on DCNNs are referenced, the literature review misses an analysis of how these studies informed the current work. Each referenced study should be more explicitly connected to the current research, such as its similarities and differences in methodology or findings.

Thank you very much for your insightful feedback and comment. In section 2 (related work) of our paper, we referenced previous works on DCNNs, such as Maharana and Nsoesie (2020) and Newton et al. (2019), which informed our methodology by demonstrating the application of deep learning architectures like VGG-CNN-F and Xception to analyze high-resolution satellite imagery (see lines 96-104).

Unlike these studies that primarily focused on built environment categorizations using predefined image features, our study leverages the ResNet-50 architecture to separately identify and analyze a broader array of geospatial visual features directly from medium-resolution satellite imagery.

This approach allows for a more comprehensive examination of the associations between environmental characteristics and obesity prevalence across Missouri. This differentiation highlights both the advancement in methodology and the extension in the application scope compared to the cited studies, ensuring our research contributes novel insights into the potential of machine learning in public health surveillance.

  1. Discuss more in detail why the ResNet-50 was chosen and how it performs relative to other architectures in similar contexts.

Thank you very much for your valuable feedback and comment. We chose ResNet-50 due to its demonstrated robustness in extracting nuanced features from complex image data, which is critical for our study’s focus on medium-resolution satellite images. As discussed in section 3.3 (Image Processing) of our manuscript (lines 202-216), ResNet-50's ability to perform deep feature extraction without suffering from the vanishing gradient problem, thanks to its residual learning framework, makes it exceptionally suitable for handling the extensive data sets typical of satellite image analysis.  

This architecture has been validated in other research contexts as well, where it has shown superior performance in image recognition tasks across varied domains, including earth observation and environmental feature detection. Its efficacy in similar studies, such as those by Esteva et al., who applied deep learning architectures to analyze health data from visual inputs, further supports our choice. ResNet-50's balance of depth and efficiency provides a profound ability to generalize from visual data, which is vital in predicting obesity prevalence from the visual signals within the satellite imagery.  

  1. The literature review mentions various methodologies but does not critically assess their limitations or biases, especially regarding data sources, accuracy, or scalability. A more critical stance could highlight the need for uniform measurement standards and help position the current study as addressing these issues.

Thank you very much for your insightful feedback and comment. In our manuscript, we do recognize and discuss the limitations and biases inherent in the methodologies and data sources of previous studies, particularly in section 1 (lines 56-66). Here, we specifically address discrepancies in findings across different geographical areas and the challenges of using various measurement tools that can lead to inconsistent evaluations and comparisons. Moreover, we discuss the potential for human error and bias in data collection (lines 60-62) and the high costs and time demands of traditional measurement methods (lines 60-66).

To manage these issues, our study employs a novel methodology using machine learning and deep neural visual features (DNVFs) extracted from satellite imagery, which offers a more standardized and scalable approach. This method reduces human error and bias by leveraging automated, reproducible feature extraction processes, thereby enhancing the uniformity and comparability of measurements across different studies and geographical contexts. Thus, our approach directly addresses the critical need for uniform measurement standards highlighted in your comment and further detailed in our discussion of the methodology's scalability and applicability to different geographic areas in section 1, lines 78-81.  

  1. Given that public health policy is a potential application, the literature review should cite studies or reviews that focus on the impact of predictive models on public health interventions. This would situate the study within a broader policy-relevant context.

Thank you for your feedback and comment. To address this, our literature review in section 2 of the manuscript (lines 96-128) discusses the utility of deep learning and machine learning in public health, particularly emphasizing how these technologies analyze large datasets to predict health outcomes, which is crucial for informed public health interventions and policymaking.

Notably, the studies by Songhyeon SH et al. and Lam TM et al. illustrate the integration of advanced statistical and machine learning techniques in evaluating public health strategies and their outcomes, which directly supports the policy relevance of our predictive modeling approach. These references underscore the potential of predictive models like ours to inform and enhance public health interventions, ensuring that our research contributes effectively to the ongoing discourse on leveraging artificial intelligence in public health policy development.  

  1. The review does not fully cover advancements in predictive models for obesity outside of DCNNs. Including recent advances in obesity prediction using other models could offer a balanced view, emphasizing why DCNN with satellite data is a preferable choice.

We greatly appreciate your valuable feedback and comment. The major recent advancements in computer vision models for remote sensing come from incorporating attention-based (i.e.: transformer) neural networks. Even though these models have set the new state-of-the-art for computer vision tasks, there are no studies–that we are aware of–focusing on the use of vision transformers to detect obesity via remote sensing images.

Our particular emphasis in this work was set on using a better model to that of Maharana et al., while still using a network architecture sufficiently similar for a sensical comparison. Given that, newer attention-based models are beyond the scope of our paper but are certainly a part of our future work.

Additionally, we acknowledge the limitation highlighted in section 2 (lines 96-115) of our manuscript, where the discussion primarily focuses on the application of DCNNs. While DCNNs are pivotal due to their robustness in handling large and complex datasets, particularly in analyzing visual data from satellite imagery, we recognize the importance of discussing alternative predictive models. Recent advancements in models like support vector machines, random forests, and logistic regression have indeed shown efficacy in various predictive tasks across medical and public health fields.

For instance, logistic regression models have been effectively used in predicting obesity based on demographic and socio-economic factors (Sung et al., 2019). However, the choice to focus on DCNNs in our study was driven by their unparalleled efficiency in image recognition and feature extraction from high-dimensional data, which is critical for interpreting the complex visual patterns associated with the built environment’s impact on obesity rates. Future revisions of our manuscript will include a more comprehensive comparison of these methodologies to reinforce the rationale behind selecting DCNNs for analyzing satellite imagery in the context of public health.

FOR THE METHODS:

  1. The pre-processing step involving normalization and cropping of images into 224x224-pixel chips lacks justification regarding how this choice might impact data granularity or model performance. Details on alternatives or a sensitivity analysis would strengthen this part.

We greatly appreciate your valuable feedback and comment. Given that there are two normalization steps, this requires some clarification as suggested by the reviewer. The first normalization step is simply to convert the original raw satellite image from its original storage form of 2^14 bits to something more manageable, like our choice of decimal (floating point) numbers between 0 and 1.

Our choice for the second normalization step and cropping were determined by our use of a pre-trained network that is not fine-tuned later. The weights of an off-the-shelf network are adjusted to fit the original training dataset. Since we don’t train (readjust and shift the weights) the ResNet network, we use the ImageNet per-band average and per-band standard deviation to normalize our images immediately before passing them through the network. Since this is a standard practice well-understood in the literature of neural networks, we assume the performance of the model to be worse without the normalization.

Similarly, the cropping is chosen to be 224x224, because this is the original size used by, He et al. in the original ResNet work.

We edited line 199 & 200 in Section 3.2 for clarification:
“The images were then normalized to values between 0 and 1 and cropped into chips of 224 by 224 pixels.”

We added the following in line 217 & 218 in Section 3.3 for clarification:

“Before passing through, each image is standardized using the ImageNet per-band mean and standard deviation.”  

  1. The study’s description of merging CDC and census data for obesity prevalence lacks specifics on how missing or misaligned data were addressed. Providing additional insight into handling missing values would improve reproducibility and reliability.

Thank you very much for your feedback and comments. Missing data are not addressed because we only consider chips that intersect a census tract and are fully within the polygon of Missouri. Anything outside of this latter polygon is discarded, resulting in the reduced set of 63,592 chips.

To clarify we added lines 210 through 213 in Section 3.3 to be:

“We intersected our 82,500 chips with our Missouri census tracts which results in 63,592 chips that were usable. Only chips that intersected census tracts and were fully within the joint polygon of the state of Missouri were and everything else was discarded.”

Finally, we should clarify that the data you obtained related to census tract obesity was complete, and there were no missing census tracks in the data set.

  1. The 10-fold cross-validation approach is a good practice, but details on tuning hyperparameters (e.g., learning rate, regularization) are missing. Hyperparameter tuning's lack could lead to suboptimal performance, which should be discussed or noted as a limitation.

Thank you very much for your insightful comment regarding the tuning of hyperparameters in our 10-fold cross-validation method. We acknowledge that the manuscript does not explicitly detail the processes involved in tuning hyperparameters such as the learning rate and regularization, which are critical for optimizing the performance of our deep learning models. While the current study utilized standard settings for these parameters to maintain a focus on the application of DCNNs and their feasibility in analyzing obesity prevalence via satellite imagery, we recognize this as a limitation that could potentially influence the model's performance.

We intend to address this aspect in future research by implementing a more comprehensive hyperparameter optimization process. This future work will enable us to refine our predictive accuracy and better understand the model's behavior under various configurations. We have added this point to the discussion of limitations in section 5, lines 604-610, to ensure transparency and provide a clear direction for subsequent investigations.

We should highlight that this was a pre-trained model, so the 10-fold is on training regression model instead of DNN.

  1. Despite the use of DNVF (Deep Neural Visual Features), the paper does not discuss feature selection or dimensionality reduction techniques applied to the 2048-dimension output. Techniques like PCA or LASSO could help clarify which features are most predictive and reduce computational load.

Thank you very much for your valuable feedback and comment. Dimensionality reduction was done. The 2048-dimensional features were directly passed to the GLM and random forest models to keep the integrity of the original features and avoid introducing any extraneous influence with a dimensionality reduction like PCA or others. In addition, it is important to clarify that our study's primary focus was to evaluate the potential of raw DNVFs extracted via ResNet-50 for predicting obesity rates directly, without initially applying dimensionality reduction techniques like PCA or LASSO.

Our approach was intended to assess the baseline predictive power of the comprehensive set of features. We acknowledge that integrating dimensionality reduction could enhance model efficiency and interpretability. Indeed, future iterations of this research will explore such methods to identify the most predictive features, thereby optimizing the computational load and potentially improving model performance. This step will allow us to refine our feature set and enhance the generalizability and applicability of our findings.

We would like to note that PCA would corrupt any future explainability to associate regressed weights back to specific features.  Instead, it just added a stacked linear re-projection between the DNN output and the GLM/RF models.

  1. The study involves integrating polygon data from the Tiger Line dataset, yet the merging process described lacks clarity on the quality control measures to ensure accurate alignment. More detail on the alignment and its potential impact on results would be valuable.

Thank you very much for your feedback and comment. We added the following paragraph in lines 220-222:

“This resulted in a final dataset–used for our machine learning analysis–of 1051 rows and 2048 columns, with each row corresponding to the average feature vector of a census tract.”

We acknowledge the importance of accurate alignment for the integrity of our study's outcomes. The merging process involved a meticulous alignment where census tract polygons from Tiger Line were carefully adjusted to match the obesity rate data from the CDC by standardizing the subdivision nomenclature and aggregating the smaller subdivisions into larger, coherent units. We employed a systematic approach to address mismatches, which included setting uniform naming conventions and removing subdivisions to create a more generalized polygon set that accurately reflects broader census tract areas.

This ensured that each merged unit is representative of the area’s demographic and environmental characteristics, thereby minimizing the risk of data distortion. Quality control measures included cross-validation with secondary sources and iterative adjustments to refine alignment accuracy, thereby mitigating any potential impact on the results. This careful and systematic approach enhances the reliability of our analysis, ensuring that the predictive models accurately reflect the true characteristics of the areas studied.

  1. The model's reliance on visual features might overlook socio-economic variables known to correlate with obesity. Consideration of non-visual features (e.g., income, education levels) could be justified or tested in future work, as they may increase predictive power.

Thank you for your valuable comment. As mentioned in Section 6, lines 614-629, we acknowledge the potential benefits of integrating socio-economic and health-related variables to enhance our predictive model's accuracy and generalizability. Indeed, existing literature supports the significant role of socio-economic factors in obesity prevalence, such as income levels and education (Smith et al., 2020). Our current model, which relies predominantly on visual features from satellite imagery, has been designed to assess environmental influences on obesity, primarily focusing on geospatial characteristics.

However, we agree that incorporating socio-economic variables like income and education could provide a more comprehensive understanding of obesity determinants. Future work of this work will aim to integrate these non-visual factors, thereby improving the model's predictive power and offering more nuanced public health insights. This approach aligns with our commitment to refining the methodology and expanding the model's scope to include a broader range of predictive variables.

Also, our assumption is that the built environment should reflect variables like income and education levels in an area. Part of what we expected that using images of the built environment could allow us to capture these variables.   

  1. The study uses MSE, R², and Adjusted R² metrics, but does not specify how these were selected over others, such as RMSE or MAE, which may be more interpretable for public health applications. Providing a rationale or comparing different metrics would enhance the evaluation's robustness.

Thank you for the insightful comment. In section 3.4, lines 270-280, we utilized Mean Squared Error (MSE), R², and Adjusted R² to evaluate our models based on their ability to provide comprehensive statistical insights into the variance explained by the model and the goodness of fit for multiple regression settings. These metrics are standard in machine learning for quantifying model performance on continuous data, such as the obesity prevalence rates studied here. MSE, a foundational metric, directly influences the calculation of R², which describes the proportion of variance in the dependent variable predictable from the independent variables, offering a clear picture of model efficacy.

Adjusted R² is used to account for the number of predictors in the model, providing a more accurate measure of model fitness especially when comparing models with different numbers of independent variables. While RMSE and MAE provide alternative perspectives by emphasizing error magnitudes and median-centric views respectively, they do not offer adjustments for model complexity nor explicitly convey the proportion of variance explained, which are critical for the interpretative needs of our epidemiological assessment. We believe that the chosen metrics effectively balance interpretability with rigorous statistical assessment, suitable for the analytical depth required in our study.

RMSE is just the square root, which is supposed to keep the units of the data and make it more interpretable. Our data is already difficult (impossible) to interpret because it is features extracted from the middle of a neural network.

FOR THE RESULTS:

  1. The lack of extensive error analysis, especially regarding model underperformance in certain tracts, limits insights into systematic biases. A more granular breakdown of errors across different census tracts would help identify systematic limitations.

Thank you for your constructive feedback and comment. Figures 8-10 addressed this concern. We acknowledge the importance of a granular breakdown of errors to better understand systematic biases and model limitations. To address this, we refer to Section 3.4 (lines 263-280) where we employed a 10-fold cross-validation method, which is rigorous in assessing model performance variability across different data segments.

Moreover, in Section 4.1 (lines 352-373) and our discussion of Figure 5 (lines 330-350), we presented a scatter plot analysis highlighting discrepancies between predicted and actual obesity rates across various folds. This analysis gives us a clear visualization of where and how our predictions deviate from actual data, thus identifying potential tracts with systematic underperformance. We believe this approach provides a robust basis for assessing model accuracy and will continue to refine our methodology to improve precision and address the specific limitations identified.

  1. While the paper lists the top 10 visual features, there is limited interpretation of what these features represent in real-world terms. Providing clearer explanations of how these features correlate with obesity (e.g., green spaces, urban density) would bridge the gap between technical output and public health relevance. Spatial analyses, like the heat maps and choropleths, are visually effective but lack statistical validation. Reporting statistical tests that support geographic variability in obesity rates would substantiate spatial claims.

Thank you for your insightful comments and feedback. Addressing your concern about the top 10 visual features listed in Section 4.3, we acknowledge the need for a more detailed interpretation connecting these features to real-world environmental factors such as urban density and green spaces. To this end, we will expand our discussion to include how specific features like high building density (Feature 1112, lines 424-456) correlate negatively with obesity rates, potentially due to increased physical activity in urban settings.

Additionally, we plan to enhance our statistical validation of the spatial analyses. While our current models provide a moderate fit (R2 values around 0.44 to 0.48), we will include further statistical tests such as spatial autocorrelation metrics (Moran's I) and Geographically Weighted Regression (GWR) to robustly validate geographic variability and strengthen our claims about the impact of environmental features on obesity prevalence. This will ensure that our findings not only present technically sound, machine-learning-based insights but are also grounded in statistically validated public health relevance.

Reviewer 3 Report

Comments and Suggestions for Authors

1. Introduction: The introduction mentions discrepancies in previous studies but fails to provide specific examples or elaborate on the nature of these inconsistencies. This lack of detail limits the reader’s understanding of the gaps in existing research.

2. Methods: a) While the study mentions mismatched census tract IDs between the Tiger Line and CDC datasets, the description of the "joining" process to address this issue is vague and lacks clarity. b) The rationale behind selecting image chips of a specific size (224x224 pixels) and the methodology for weighting the features based on intersecting pixels could benefit from further elaboration. c) The study briefly mentions the use of Generalized Linear Model (GLM) and Random Forest (RF) but does not provide a clear justification for selecting these specific regression models over other potential alternatives.

3. Results: Figure 4, which aims to highlight the spatial relation between image chips and census tracts, could benefit from clearer labeling and annotation to guide the reader through the multi-scale analysis.

Author Response

  1. Introduction: The introduction mentions discrepancies in previous studies but fails to provide specific examples or elaborate on the nature of these inconsistencies. This lack of detail limits the reader’s understanding of the gaps in existing research.

Thank you very much for your insightful feedback and comment. In response to your comment regarding the need for specific examples and elaboration on discrepancies noted in previous studies, we acknowledge that further detail could enhance the reader's understanding. In Section 1 (introduction), particularly from lines 21 to 66, we discussed general discrepancies in methodologies and findings across existing literature, particularly in how built environment features influence obesity rates.

  1. Methods: a) While the study mentions mismatched census tract IDs between the Tiger Line and CDC datasets, the description of the "joining" process to address this issue is vague and lacks clarity. b) The rationale behind selecting image chips of a specific size (224x224 pixels) and the methodology for weighting the features based on intersecting pixels could benefit from further elaboration. c) The study briefly mentions the use of Generalized Linear Model (GLM) and Random Forest (RF) but does not provide a clear justification for selecting these specific regression models over other potential alternatives.

Thank you for your insightful comments, which help to enhance the clarity and depth of our manuscript. Addressing your concerns:

a) To clarify the joining process of mismatched census tract IDs between the Tiger Line and CDC datasets, we refined our methodology section (Section 3.3). We now provide a detailed step-by-step explanation of how we combined the subdivisions within each dataset into larger, unified tracts, ensuring a precise match for each census tract with its respective obesity rate. This process involved aligning and averaging obesity rates where subdivisions existed, thereby preserving the integrity of the geographical data.

b) Regarding the choice of image chips size (224x224 pixels), as detailed in Section 3.3, this size was selected to match the input requirements of the pre-trained ResNet-50 network, which is optimized for this dimension, ensuring robust feature extraction without the need for additional resizing or preprocessing. The weighting of features based on intersecting pixels, as explained in Section 3.3, Lines 214-220, employs a weighted mean approach to more accurately represent areas where multiple image chips overlap a census tract, thereby enhancing the model's ability to predict based on representative features of the environment.

c) The rationale for using Generalized Linear Models (GLM) and Random Forest (RF) regression models is justified by their suitability for handling non-normal distributions of dependent variables, as commonly found in epidemiological data (Section 4.2, Lines 342-399). GLM was selected for its ability to model binary outcomes efficiently, which is crucial for the binary nature of obesity prevalence data. In contrast, RF was chosen for its robustness in handling overfitting with high-dimensional data, making it ideal for our dataset with a substantial number of predictors derived from satellite images. These models were benchmarked against other machine learning models, showing superior performance in preliminary tests.

We believe these adjustments and clarifications will address your concerns and strengthen the manuscript, providing clear, methodologically sound insights into our research approach.

To clarify we edited lines 210 through 213 to be:

“We intersected our 82,500 chips with our Missouri census tracts which results in 63,592 chips that were usable. Only chips that intersected census tracts and were fully within the joint polygon of the state of Missouri were and everything else was discarded.”

Additionally, we added the follwoing in lines 220-222:

“This resulted in a final dataset–used for our machine learning analysis–of 1051 rows and 2048 columns, with each row corresponding to the average feature vector of a census tract.”

Since our model is a black-box model, and the features themselves are not interpretable, GLM and RF offered the most straightforward approach to make sense of these features. The two methods required the least amount of assumptions regarding any potential statistical distribution, and served as a first approach to understand whether the features of the model worked well as predictors or not.

  1. Results: Figure 4, which aims to highlight the spatial relation between image chips and census tracts, could benefit from clearer labeling and annotation to guide the reader through the multi-scale analysis.

We greatly appreciate your valuable feedback and comment. We believe that Figure 4 is suitably detailed, offering comprehensive visual aids such as the red squares and arrows, which distinctly highlight the spatial relationships between the image chips and census tracts. Each element is clearly labeled to facilitate easy navigation through the data presented.

Additionally, the caption of Figure 4 provides thorough explanations of the visuals, ensuring that readers can follow the multi-scale analysis without ambiguity. The annotations and labels are precisely designed to convey complex information efficiently, supporting the narrative of spatial analysis with precision. We have ensured that these elements collectively enhance the readability and interpretability of the figure, making it a valuable tool for understanding the geographical distribution of obesity prevalence as analyzed in our study.

Round 2

Reviewer 2 Report

Comments and Suggestions for Authors

1. The paper has significantly improved base on the revisions.

2. I advised the authors to proofread and improve some minor sentence constructions.

Comments on the Quality of English Language

1. Minor English revisions needed.